# An Epidemic Model with Infection Age and Vaccination Age Structure

**Glenn Webb [1],* and Xinyue Evelyn Zhao [2],***

1   Department of Mathematics, Vanderbilt University, Nashville, TN 37240, USA
2   Department of Mathematics, University of Tennessee, Knoxville, TN 37996, USA
*   glenn.f.webb@vanderbilt.edu (G.W.); xzhao45@utk.edu (X.E.Z.)

**Abstract:** A model of epidemic dynamics is developed that incorporates continuous variables for infection age and vaccination age. The model analyzes pre-symptomatic and symptomatic periods of an infected individual in terms of infection age. This property is shown to be of major importance in the severity of the epidemic, when the infectious period of an infected individual precedes the symptomatic period. The model also analyzes the efficacy of vaccination in terms of vaccination age. The immunity to infection of vaccinated individuals varies with vaccination age and is also of major significance in the severity of the epidemic. Application of the model to the 2003 SARS epidemic in Taiwan and the COVID-19 epidemic in New York provides insights into the dynamics of these diseases. It is shown that the SARS outbreak was effectively contained due to the complete overlap of infectious and symptomatic periods, allowing for the timely isolation of affected individuals. In contrast, the pre-symptomatic spread of COVID-19 in New York led to a rapid, uncontrolled epidemic. These findings underscore the critical importance of the pre-symptomatic infectious period and the vaccination strategies in influencing the dynamics of an epidemic.

**Keywords:** COVID-19; data; transmission; asymptomatic; symptomatic; vaccination

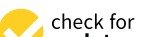



## 1. Introduction

The objective of this work is to model the effects of quarantine, vaccination, and hospital isolation on the transmission of an infectious disease in an epidemic population of susceptible individuals and infected individuals. The focus of the model is on the infectious and symptomatic periods of an infective, which may or may not coincide. For a viral respiratory disease with severe morbidity and mortality, the symptomatic period typically results in hospital isolation as soon as the disease is recognized as a major public health problem. If the infectious and symptomatic periods coincide, or if the infectious period follows the appearance of symptoms, then the hospitalization of symptomatic patients is an effective method of isolating infectious individuals, reducing the potential for disease transmission to others. If, however, the infectious period precedes the symptomatic phase, then there is much greater potential for disease transmission to those who are susceptible. The efficacy of vaccination during the epidemic is incorporated into the model to account for the acquisition of immunity over a time period of vaccinated individuals.

The model is applicable to influenza epidemics such as the SARS epidemic of 2003 and the current COVID-19 pandemic. The 2003 SARS epidemic was contained, in part, because SARS infectives were infectious after manifesting symptoms, which allowed for their identification and controlled isolation in hospitals. Vaccination has played a key role in the containment of the current COVID-19 pandemic. The central point of the study here is that in a future epidemic comparable to the 2003 SARS epidemic and the current COVID-19 pandemic, quarantine, vaccination, and hospital isolation will be critical elements of containment.

The literature on epidemic models is extensive, with structure variables and vaccination elements. In our References, we have listed many of such works [1–110]. The collection

of works in our References provides a useful resource of research contributions to the subject of our work here.

The organization of this paper is as follows: In Section 2.1, we present the compartments and parameters of the model. In Section 2.2, we present the equations of the model. In Section 2.3, we analyze the model. In Section 2.4, we apply the model to the 2003 SARS epidemic in Taiwan. In Section 2.5, we apply the model to the COVID-19 epidemic in New York. In Section 3, we provide a discussion of our results and highlight some future work.

## 2. Materials and Methods

### 2.1. State Variables of the Model

The state variables of the model are $S(t)$ = susceptible individuals at time $t$, $V(t)$ = vaccinated individuals at time $t$, $E(t)$ = exposed individuals at time $t$ (those who have been infected but are not yet infectious), $I(t)$ = infective individuals at time $t$ (those capable of transmitting the disease), $H(t)$ = hospitalized infectives at time $t$ (including mortality), $Q(t)$ = quarantined infectives at time $t$, and $R(t)$ = recovered infectives at time $t$. The interactions among these compartments are depicted in Figure 1.

The key features of this model are (1) infected individuals are tracked by disease age $a_i$, and the incubation, infectious, and symptomatic stages of the disease are modeled by the disease age of the infected individual, and (2) vaccinated individuals are tracked by vaccination age $a_v$, and their susceptibility to infection depends on their vaccination age as they gradually acquire and lose immunity.

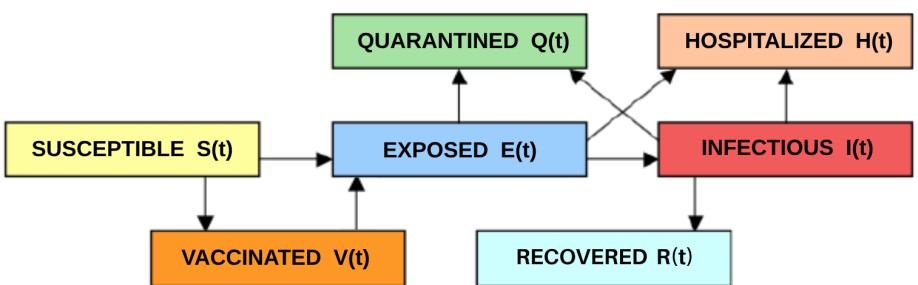

**Figure 1.** Diagram of susceptible, vaccinated, exposed, infectious, hospitalized, quarantined, and recovered compartments of the model and the interactions of these compartments in the equations of the model.

The infected population has infection age density $i(a_i, t)$. Infectives begin the disease course at age $a_i = 0$, are infected but noninfectious (exposed) from age $a_i = 0$ to age $a_i = r$, and infectious from age $a_i = r$ to age $a_i = r + s$. Infectives are no longer infectious after reaching the disease age $a_i = r + s$ and are considered recovered, with the assumption that they cannot be re-infected. Thus:

$$E(t) = \int_0^r i(a_i, t) da_i, \tag{1}$$

$$I(t) = \int_r^{r+s} i(a_i, t) da_i, \tag{2}$$

$$R(t) = \int_0^t i(r + s, \hat{t}) d\hat{t}, t \geq 0. \tag{3}$$

Infectives with infection age $a_i \leq r + s$ can be removed from the exposed class $E(t)$ or the infectious class $I(t)$ at time $t$ due to hospitalization or quarantine. Mortality due to the disease is included in the hospitalized compartment. Transmission of infection to susceptibles, hospitalization, manifestation of symptoms, and quarantine all depend on disease age. It is also assumed that hospitalized, quarantined, and recovered infectives do not transmit the disease to susceptibles.

The vaccinated population has vaccination age density $v(a_v, t)$. Vaccinated individuals begin with vaccination age $a_v = 0$ and then have increasing or decreasing immunity to infection as their vaccination age $a_v$ increases over time. We assume there are no vaccinated individuals at $t = 0$, and the vaccination starts on or after $t = 0$, thus $a_v \leq t$. The number of vaccinated individuals at time $t$ is

$$V(t) = \int_0^t v(a_v, t) da_v. \tag{4}$$

The population of vaccinated individuals has a gain from the susceptible class and a loss to the infected class, since vaccination efficacy is incomplete.

The model does not take into account demographics (births and deaths) of the population. The time scale of the model (the units of $t$ are typically days) is comparable to a small fraction of the lifespan of individuals in the population. The asymptotic behavior of the model populations, corresponding to large time, is comparable to a small fraction of the typical lifespan of individuals. For human populations, the typical time units are days and the meaningful time scale of the model is several years.

The parameters of the model are as follows: $\alpha(a_i)$ is the disease transmission rate from an infectious individual with infection age $a_i$ to a susceptible individual, $\nu$ is the rate of vaccination of susceptibles, $1 - \sigma(a_v)$ measures the effectiveness of vaccination for a vaccinated individual with vaccination age $a_v$, and $\beta_H(a_i)$ and $\beta_Q(a_i)$ are the transition rates of infectives with infection age $a_i$ to hospitalization and quarantine, respectively.

### 2.2. Equations of the Model

The equations of the model are as follows: for $t \geq 0$,

$$\frac{d}{dt} S(t) = -\left( \int_r^{r+s} \alpha(a_i) i(a_i, t) da_i + \nu \right) S(t), \tag{5}$$

$$\frac{\partial}{\partial t} i(a_i, t) + \frac{\partial}{\partial a_i} i(a_i, t) = -\left( \beta_H(a_i) + \beta_Q(a_i) \right) i(a_i, t), 0 \leq a_i \leq r + s, \tag{6}$$

$$i(0, t) = \int_r^{r+s} \alpha(a_i) i(a_i, t) da_i \left( S(t) + \int_0^t \sigma(a_v) v(a_v, t) da_v \right), \tag{7}$$

$$\frac{\partial}{\partial t} v(a_v, t) + \frac{\partial}{\partial a_v} v(a_v, t) = -\sigma(a_v) \left( \int_r^{r+s} \alpha(a_i) i(a_i, t) da_i \right) v(a_v, t), 0 \leq a_v \leq t, \tag{8}$$

$$v(0, t) = \nu S(t), \tag{9}$$

with initial conditions

$$S(0) = S_0, i(a_i, 0) = i_0(a_i), v(a_v, 0) = v_0(a_v) \equiv 0. \tag{10}$$

(we assume there are no vaccinated individuals at $t = 0$).

### 2.3. Analysis of the Model

Assume the following hypothesis: $\nu \geq 0$, $\alpha$ is non-negative and piecewise continuous on $[r, r+s)$; $\beta_H$ and $\beta_Q$ are non-negative and piecewise continuous on $[0, r+s)$; $\sigma$ is non-negative and piecewise continuous on $[0, \infty)$; $S_0 > 0$, $i_0$ is non-negative and piecewise continuous on $[0, r+s)$; and $v_0 \equiv 0$ on $[0, \infty)$. The existence of unique non-negative solutions in $[0, \infty) \times L^1[0, r+s) \times L^1[0, \infty)$ to the system of Equations (5)–(9), with initial conditions (10), can be proven with the techniques developed in [111]. The asymptotic behavior of this system without vaccination is investigated in [112,113]. We prove the following asymptotic behavior of the solutions with vaccination by the below theorem.

**Theorem 1.** *Assume that for $0 \le a_i \le r + s$, $\beta_H(a_i) + \beta_Q(a_i) \ge \bar{\beta} > 0$, $0 \le \alpha(a_i) \le \bar{\alpha} > 0$, and $0 \le \sigma(a_i) \le \bar{\sigma} > 0$. The solutions of (5)–(9) with initial conditions (10) have the following asymptotic behavior:*

$$\lim_{t \to \infty} S(t) = S_\infty \ge 0, \lim_{t \to \infty} E(t) = 0, \lim_{t \to \infty} I(t) = 0. \tag{11}$$

*If $\nu > 0$ (vaccination), then $S_\infty = 0$. If $\nu = 0$ (no vaccination), then $S_\infty$ satisfies*

$$S_\infty = exp\left[ -\left( \Gamma + (S(0) - S_\infty)\Lambda \right) \right] S(0), \tag{12}$$

*where*

$$\Gamma = \int_r^{r+s} \alpha(a_i) \int_0^{a_i} i_0(u) exp\left[ -\int_u^{a_i} (\beta_H(b) + \beta_Q(b))db \right] du \, da_i, \tag{13}$$

$$\Lambda = \int_r^{r+s} \alpha(a_i) exp\left[ -\int_0^{a_i} (\beta_H(b) + \beta_Q(b))db \right] da_i. \tag{14}$$

**Proof.** Let $\beta(a_i) = \beta_H(a_i) + \beta_H(a_i) \ge \bar{\beta} > 0, 0 \le a_i \le r + s$. We first prove (11). From (5) for $t \ge 0$:

$$S(t) \;=\; exp(-\nu t) exp\left[ -\int_0^t \left( \int_r^{r+s} \alpha(a_i) i(a_i, \hat{t}) da_i \right) d\hat{t} \right] S(0) \tag{15}$$

Then, it implies that $S(t)$ is non-increasing and $S_\infty = \lim_{t \to \infty} S(t) \ge 0$. It is also clear that $S_\infty = 0$ if $\nu > 0$. In addition, by evaluating the integral of (5) from 0 to $t$, we have

$$S(t) - S(0) = -\int_0^t \left( \int_r^{r+s} \alpha(a_i) i(a_i, \hat{t}) da_i \right) S(\hat{t}) d\hat{t} - \nu \int_0^t S(\hat{t}) d\hat{t},$$

which is equivalent to

$$S(t) + \int_0^t \left( \int_r^{r+s} \alpha(a_i) i(a_i, \hat{t}) da_i \right) S(\hat{t}) d\hat{t} = S(0) - \nu \int_0^t S(\hat{t}) d\hat{t}. \tag{16}$$

We then derive equations for $V(t)$. Combining (4), (8), and (9), we obtain, for $t \ge 0$,

$$
\begin{aligned}
V'(t) &= \frac{d}{dt}\left( \int_0^t v(a_v, t) da_v \right) = v(t, t) + \int_0^t v_t(a_v, t) da_v \\
&= v(t, t) + \int_0^t \left[ -v_{a_v}(a_v, t) - \sigma(a_v)\left( \int_r^{r+s} \alpha(a_i) i(a_i, t) da_i \right) v(a_v, t) \right] da_v \\
&= v(t, t) - v(t, t) + v(0, t) - \int_0^t \sigma(a_v)\left( \int_r^{r+s} \alpha(a_i) i(a_i, t) da_i \right) v(a_v, t) da_v \\
&= \nu S(t) - \int_0^t \left( \int_r^{r+s} \alpha(a_i) i(a_i, t) da_i \right) \sigma(a_v) v(a_v, t) da_v,
\end{aligned}
$$

which integrates to

$$
\begin{aligned}
V(t) + \int_0^t \left( \int_0^{\hat{t}} \left[ \left( \int_r^{r+s} \alpha(a_i) i(a_i, \hat{t}) da_i \right) \sigma(a_v) v(a_v, \hat{t}) \right] da_v \right) d\hat{t} \\
= V(0) + \int_0^t \nu S(\hat{t}) d\hat{t}.
\end{aligned}
\tag{17}
$$

Then, (15), (17), and the non-negativity of solutions imply that $\lim_{t\to\infty} V(t)$ exists. Next, we consider $E(t)$ and $I(t)$. Using Equations (1), (2), and (6), we have, for $t \geq 0$,

$$
\begin{aligned}
E'(t) &= \frac{d}{dt}\left(\int_0^r i(a_i, t)da_i\right) = \int_0^r i_t(a_i, t)da_i = \int_0^r \left(-i_{a_i}(a_i, t) - \beta(a_i)i(a_i, t)\right)da_i \\
&= i(0, t) - i(r, t) - \int_0^r \beta(a_i)i(a_i, t)da_i,
\end{aligned}
\tag{18}
$$

$$
\begin{aligned}
I'(t) &= \frac{d}{dt}\left(\int_r^{r+s} i(a_i, t)da_i\right) = \int_r^{r+s} i_t(a_i, t)da_i = \int_r^{r+s} \left(-i_{a_i}(a_i, t) - \beta(a_i)i(a_i, t)\right)da_i \\
&= i(r, t) - i(r+s, t) - \int_r^{r+s} \beta(a_i)i(a_i, t)da_i,
\end{aligned}
\tag{19}
$$

where the boundary condition $i(0, t)$ is defined in (7). Adding up these two equations results in

$$
E'(t) + I'(t) = i(0, t) - i(r+s, t) - \int_0^{r+s} \beta(a_i)i(a_i, t)da_i,
$$

which integrates to

$$
E(t) + I(t) = E(0) + I(0) + \int_0^t \left(i(0, \hat{t}) - i(r+s, \hat{t}) - \int_0^{r+s} \beta(a_i)i(a_i, \hat{t})da_i\right)d\hat{t}.
\tag{20}
$$

In Equation (20), we use (7) to derive

$$
\begin{aligned}
\int_0^t i(0, \hat{t}) &= \int_0^t \int_r^{r+s} \alpha(a_i)i(a_i, \hat{t})da_i\left(S(\hat{t}) + \int_0^{\hat{t}} \sigma(a_v)v(a_v, \hat{t})da_v\right)d\hat{t} \\
&= \int_0^t \left(\int_r^{r+s} \alpha(a_i)i(a_i, \hat{t})da_i\right)S(\hat{t})d\hat{t} \\
&\quad + \int_0^t \left(\int_r^{r+s} \alpha(a_i)i(a_i, \hat{t})da_i\right)\left(\int_0^{\hat{t}} \sigma(a_v)v(a_v, \hat{t})da_v\right)d\hat{t} \\
&= \int_0^t \left(\int_r^{r+s} \alpha(a_i)i(a_i, \hat{t})da_i\right)S(\hat{t})d\hat{t} \\
&\quad + \int_0^t \left(\int_0^{\hat{t}} \left[\left(\int_r^{r+s} \alpha(a_i)i(a_i, \hat{t})da_i\right)\sigma(a_v)v(a_v, \hat{t})\right]da_v\right)d\hat{t}.
\end{aligned}
\tag{21}
$$

We observe that the first term in (21) is equal to the second term in the left-hand side of (16), and the second term in (21) equals the second term in the left-hand side of (17). Therefore, when summing up Equations (16), (17) and (20), we find that the two terms in (21) cancel each other out. Consequently, for $t \geq 0$, we have

$$
\begin{aligned}
S(t) + V(t) + E(t) + I(t) + \int_0^t i(r+s, \hat{t})d\hat{t} + \int_0^t \left(\int_0^{r+s} \beta(a_i)i(a_i, \hat{t})da_i\right)d\hat{t} \\
= S(0) + V(0) + E(0) + I(0).
\end{aligned}
\tag{22}
$$

Since $\beta(a_i) \geq \bar{\beta}$ for $0 \leq a_i \leq r+s$, (22) implies

$$
\int_0^\infty \left(E(t) + I(t)\right)dt < \infty.
\tag{23}
$$

Thus, (15), (17), (22), and (23) imply

$$
\lim_{t\to\infty} E(t) = \lim_{t\to\infty} I(t) = 0.
$$

Lastly, we prove that if $\nu = 0$ (no vaccination), then $S_\infty > 0$ and satisfies (12). From (6):

$$i(a_i, t) = \begin{cases} i_0(a_i - t)exp[-\int_{a_i-t}^{a_i} \beta(b)db], & a_i > t; \\ i(0, t - a_i)exp[-\int_0^{a_i} \beta(b)db], & a_i \leq t. \end{cases} \quad (24)$$

From (5), (7), and (24) with $\nu = 0$, for $t \geq 0$,

$$S'(t) = -i(0,t) \implies S(0) - S_\infty = \int_0^\infty i(0,t)dt. \quad (25)$$

For $t \geq 0$, (24) and (25) imply

$$\int_0^\infty \int_r^{r+s} \alpha(a_i)i(a_i, t)da_i dt$$

$$= \int_r^{r+s} \alpha(a_i)\left( \int_0^{a_i} i(a_i, t)dt + \int_{a_i}^\infty i(a_i, t)dt \right)da_i$$

$$= \int_r^{r+s} \alpha(a_i)\left( \int_0^{a_i} i_0(a_i - t)e^{-\int_{a_i-t}^{a_i} \beta(b)db}dt + \int_{a_i}^\infty i(0, t - a_i)e^{-\int_0^{a_i} \beta(b)db}dt \right)da_i$$

$$= \int_r^{r+s} \alpha(a_i)\left( -\int_{a_i}^0 i_0(u)e^{-\int_u^{a_i} \beta(b)db}du + \int_0^\infty i(0, u)e^{-\int_0^{a_i} \beta(b)db}du \right)da_i$$

$$= \int_r^{r+s} \alpha(a_i)\left( \int_0^{a_i} i_0(u)e^{-\int_u^{a_i} \beta(b)db}du + \left( \int_0^\infty i(0, u)du \right)\left( e^{-\int_0^{a_i} \beta(b)db} \right) \right)da_i$$

$$= \Gamma + (S(0) - S_\infty)\Lambda.$$

Then, (12) follows from (15). $\square$

**Remark 1.** *We claim that $|E'(t) + I'(t)|$ is bounded for $t \geq 0$, which provides a useful estimate on the values of $E'(t)$ and $I'(t)$. We shall prove this statement in two cases. For $t \geq r + s$, (7), (22), and (24) imply there exists $C_1 > 0$, such that*

$$i(r + s, t) = i(0, t - r - s)exp\left[ -\int_0^{r+s} \beta(b)db \right]$$

$$\leq \left( \int_r^{r+s} \alpha(a_i)i(a_i, t - r - s)da_i \right)$$

$$\left( S(t - r - s) + \int_0^{t-r-s} \sigma(a_i)v(a_v, t - r - s)da_v \right)$$

$$\leq \left( \bar{\alpha}I(t - r - s) \right)\left( S(t - r - s) + \bar{\sigma}V(t - r - s) \right) \leq C_1.$$

*For $t < r + s$, we again use (24) to derive*

$$i(r + s, t) = i_0(r + s - t)exp\left[ -\int_{r+s-t}^{r+s} \beta(b)db \right] \leq i_0(r + s - t).$$

*Since the initial condition $i_0$ is piecewise continuous on $[0, r + s)$, there exists $C_2 > 0$, such that*

$$i(r + s, t) \leq i_0(r + s - t) \leq C_2,$$

*for $t < r + s$.*

*Let $C^* = \max\{C_1, C_2\}$. Combining (7), (18), and (19), we have, for $t \geq 0$,*

$$\left| E'(t) + I'(t) \right| = \left| i(0,t) - i(r+s,t) - \int_0^{r+s} \beta(a_i)i(a_i,t)da_i \right|$$

$$= \left| \left( \int_r^{r+s} \alpha(a_i)i(a_i,t)da_i \right) \left( S(t) + \int_0^t \sigma(a_v)v(a_v,t)da_v \right) \right.$$

$$\left. - i(r+s,t) - \int_0^{r+s} \beta(a_i)i(a_i,t)da_i \right|$$

$$\leq \bar{\alpha}I(t)\left( S(t) + \bar{\sigma}V(t) \right) + C^* + \bar{\beta}\left( E(t) + I(t) \right).$$

*By (22), $|E'(t) + I'(t)|$ is bounded for $t \geq 0$.*

### 2.4. Application of the Model to the 2003 SARS Epidemic in Taiwan

This example is based on results in [114–116], and illustrates the case that the period of infectiousness coincides with the symptomatic period. In the SARS epidemic in Taiwan in 2003, the seriousness of the disease was recognized after an initial period, and by 24 April 2003, infected individuals were quickly identified and isolated in hospitals, with stringent control measures to prevent further disease transmission. The incubation period of SARS was from two to seven days. In this epidemic in Taiwan 2003, vaccination was not available. We first consider the model without vaccination ($v = 0$).

We assume that the incubation (exposed) period lasts from the moment of infection to day 5, and the infectious period lasts from day 5 to day 26. We assume the symptomatic period to be synonymous with the infectious period, which also lasts from day 5 to day 26 (see Scenario 1 in Figure 2). It is also assumed that, after 24 April 2003, a certain percentage of symptomatic infectives were isolated in hospitals, and gave no further transmissions to susceptibles. We will then extrapolate the model to the case where the pre-symptomatic and infectious periods overlap by one day (see Scenario 2 in Figure 2) and the case where the two periods overlap by two days (see Scenario 3 in Figure 2).

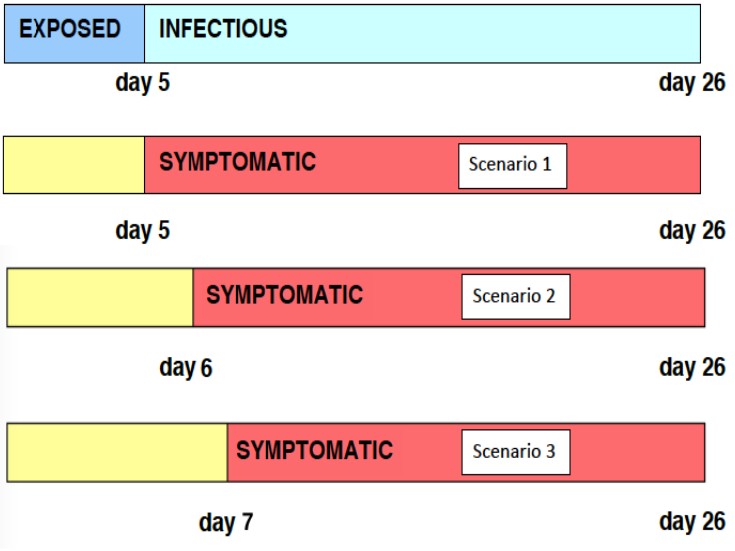

**Figure 2.** Timeline of infectious periods relative to symptom onset for SARS. The top segment displays the exposed–infectious period. Segments 2 to 4 illustrate scenarios where the infectious period coincides with the symptomatic period, precedes the symptomatic period by one day, and precedes the symptomatic period by two days, respectively.

All parameters are based on fitting the model to data [114]. The initial population of susceptibles is set at $S(0) = 6.0 \times 10^6$. It is assumed that the exposed period lasts from day 0 until day $r = 5$, and the infectious period lasts $s = 21$ days, from day $r = 5$ to day $r + s = 26$. The asymptomatic period and the exposed period coincide, as do the symptomatic period and the infectious period (see Figure 2). The transmission rate is defined as (see Figure 3):

$$\alpha(a_i) = \begin{cases} 0 & \text{if } 0 \le a_i < r, \\ 3.1 \times 10^{-8}(a_i - r) & \text{if } r \le a_i < r + 10, \\ 3.1 \times 10^{-7}\left(1.0 - \frac{a_i - r - 10}{11}\right) & \text{if } r + 10 \le a_i < r + s, \\ 0 & \text{if } r + s \le a_i. \end{cases}$$

The hospitalization rate is 54.5% per day after manifestation of symptoms at day 5 and 0.0% per day before day 5 (see Figure 3):

$$\beta_H(a_i) = \begin{cases} 0 & \text{if } 0 \le a_i < 5, \\ 0.545 & \text{if } 5 \le a_i \le 26. \end{cases}$$

We assume only pre-symptomatic infected individuals are quarantined. The quarantine rate is 2.0% per day from day 0 to day 5 and then 0.0% per day after day 5 (see Figure 3):

$$\beta_Q(a_i) = \begin{cases} 0.020 & \text{if } 0 \le a_i < 5, \\ 0 & \text{if } 5 \le a_i \le 26. \end{cases}$$

It is assumed that at time 0 the distribution of infectives $i(a_i, 0)$ is given by (see Figure 3):

$$i(a_i, 0) = \begin{cases} 12 & \text{if } a_i \le 1, \\ 5 & \text{if } 1 < a_i \le 2, \\ 19 & \text{if } 2 < a_i \le 3, \\ 9 & \text{if } 3 < a_i \le 4, \\ 15 & \text{if } 4 < a_i \le 5, \\ \frac{1}{2}(17 - a) & \text{if } 5 < a_i \le 17, \\ 0 & \text{if } 17 < a_i. \end{cases}$$

With this initial distribution $i(a, 0)$, the total number of exposed at time $t = 0$ is $E(0) = 60$ and the total number of infectious at time $t = 0$ is $I(0) = 36$. It is assumed that $H(0) = 0$, $Q(0) = 0$, and $R(0) = 0$. In Figure 4, the graphs of the exposed population $E(t)$, the infectious population $I(t)$, the cumulative number of new cases $\int_0^t i(0, \hat{t}) d\hat{t}$, and the daily number of new cases $i(0, t)$ are given and compared to the data of the epidemic from 28 April to 25 June 2003. The total number of new cases is $\approx 230$. Data for the epidemic are given in [114], with 232 cases reported for this time period.

The model can be used to evaluate the role of the susceptible size population $S(0)$ in predicting the number of cases $S(0) - S_\infty$ in the epidemic, with all other parameters and initial conditions held fixed. For this example, $S(0) = 6,000,000$ and $S_0 - S_\infty \approx 230$. In Figure 5, we use Formula (12) to plot $S(0) - S_\infty$ as a function of $S(0)$, as $S(0)$ increases from $10^5$ to $10^7$. We find that the number of cases $S(0) - S_\infty$ increases as the initial susceptible population size $S(0)$ increases.

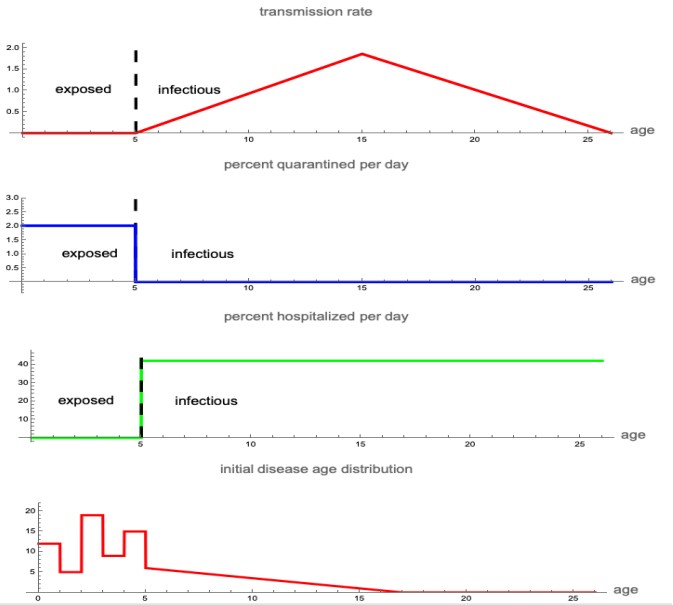

**Figure 3.** The disease age-dependent transmission rate, quarantine rate, hospitalization rate, and initial disease age distribution of infectives for the 2003 Taiwan SARS epidemic from 28 April to 5 June 2003. Vertical dashed lines separate exposed and infectious periods.

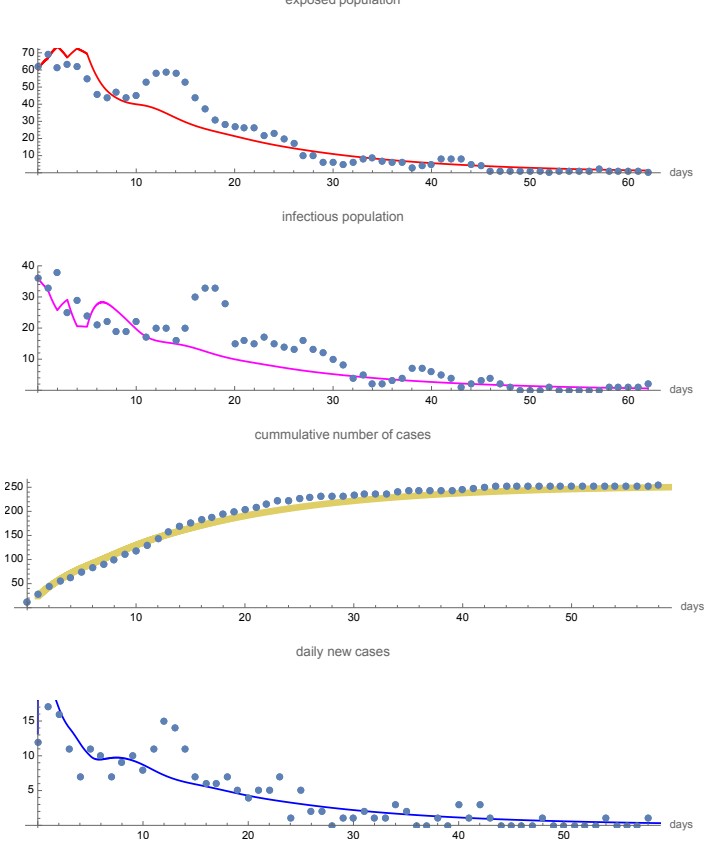

**Figure 4.** The graphs of the exposed class $E(t)$, infectious class $I(t)$, cumulative number of new cases, and daily new cases in the 2003 Taiwan SARS epidemic from 28 April to 5 June. The dotted curves are data (Taiwan Centers for Disease Control https://www.cdc.gov.tw) and the solid curves are the model simulation. $S_\infty \approx 5{,}999{,}770$. The cumulative number of cases on 5 June is approximately 230.

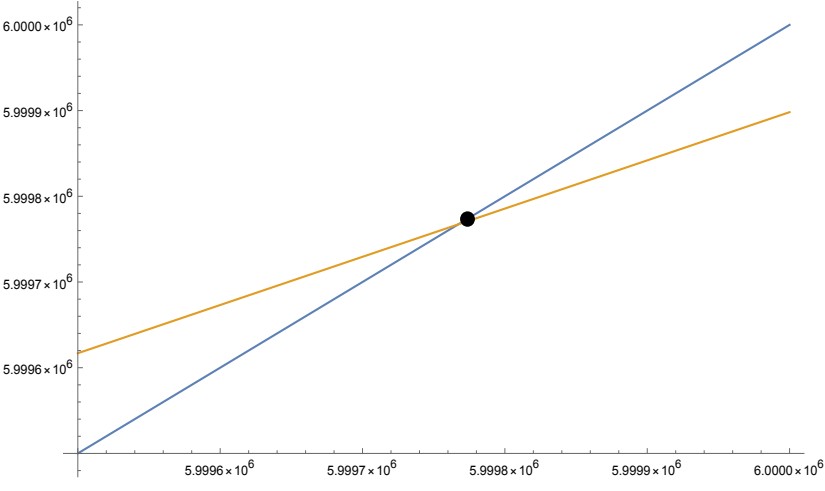

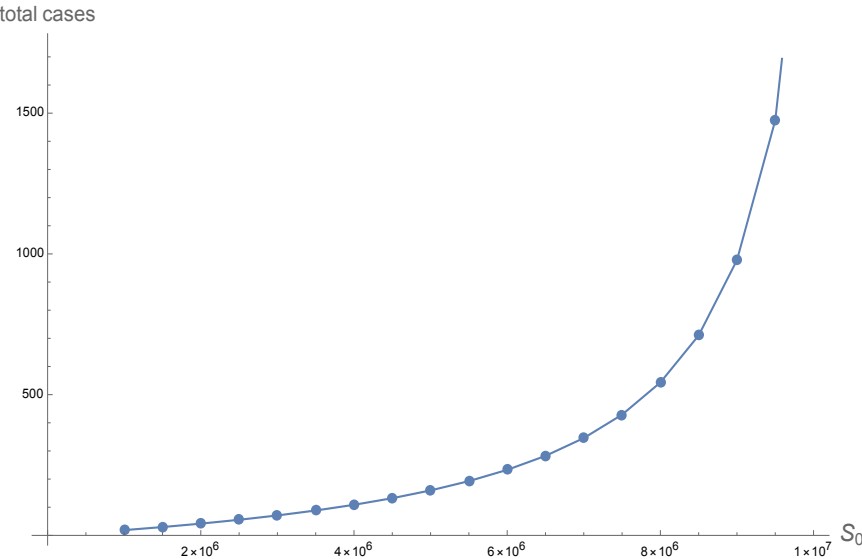

**Figure 5.** **(Top)** The blueline is the function $F(X) = X$ and the redline is the function $G(X) = exp[-(\Gamma + (S(0) - X)\Lambda)]S(0)$. The intersection of the two lines is $(S_\infty, S_\infty)$, $S_\infty \approx 5{,}999{,}770$. $\Lambda \approx 9.376 \times 10^{-8}$ and $\Gamma \approx 0.000016929$. The total number of cases is $S(0) - S_\infty \approx 230$. **(Bottom)** The number of new cases $S(0) - S_\infty$ as a function of $S(0)$, where $S_\infty$ is computed by $S_\infty = exp[-(\Gamma + (S(0) - S_\infty)\Lambda)]S(0)$. $S(0) - S_\infty$ increases as $S(0)$ increases.

**Remark 2.** *The asymptotic behavior of the solutions of (5)–(7) without vaccination ($\nu = 0$), is analogous to the asymptotic behavior of the solutions of the classic Kermack–McKendrick SEIR model [117,118]:*

$$S'(t) = -\alpha S(t)I(t), t \geq 0, \tag{26}$$

$$E'(t) = \alpha S(t)I(t) - \beta E(t), t \geq 0, \tag{27}$$

$$I'(t) = \beta E(t) - \gamma I(t), t \geq 0, \tag{28}$$

$$R'(t) = \gamma I(t), t \geq 0. \tag{29}$$

*The limiting behavior as $t \to \infty$ depends on the initial conditions $S(0), E(0), I(0)$:*

$$\lim_{t \to \infty} E(t) = 0, \lim_{t \to \infty} I(t) = 0, \lim_{t \to \infty} S(t) = S_\infty > 0,$$

*where $S_\infty$ satisfies*

$$S_\infty = S(0) + E(0) + I(0) + \frac{\gamma}{\alpha} log\left(\frac{S_\infty}{S(0)}\right). \tag{30}$$

Examples are given in Figure 6. This result is of major scientific importance because it explains why epidemic diseases, which can occur hundreds of thousands of times over evolutionary time scales, do not annihilate biological species.

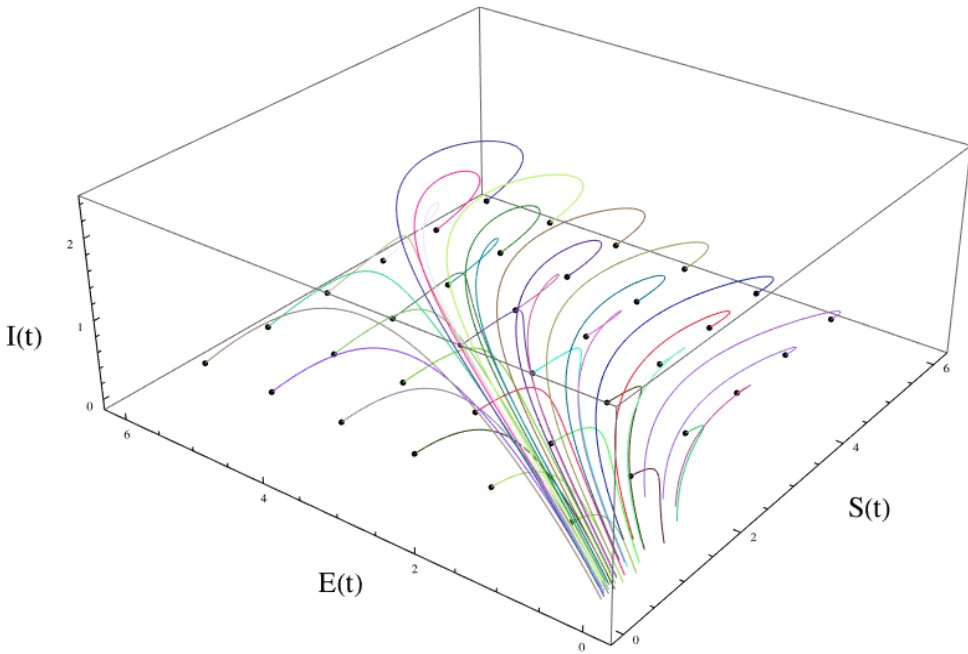

**Figure 6.** The solution $S(t), E(t), I(t)$ of the Kermack–McKendrick SEIR model (26)–(29) for varying initial values $S(0), E(0), I(0)$. The limiting behavior as $t \to \infty$ is dependent on the initial values: $\lim_{t\to\infty} E(t) = 0, \lim_{t\to\infty} I(t) = 0, \lim_{t\to\infty} S(t) = S_\infty$, where $S_\infty$ satisfies (30).

The role of hospitalization (isolation) and quarantine of infectives in the 2003 Taiwan SARS epidemic can be analyzed using the model. We consider two scenarios in which the infection period precedes the symptomatic period—the period of infectiousness begins on day 5 and the period of symptoms begins on day 6 or day 7 (see Figure 2). We also consider three scenarios in which the quarantine rate is 2.0 % per day, 4.0% per day, and 10.0% per day. We assume only pre-symptomatic infected individuals are quarantined. The parameters $\alpha$ and $\beta_H$ are as before, and $\nu = 0$ (no vaccination).

In the case that exposed infectives are symptomatic at day 6 (infectious 1 day before symptoms) and the maximum quarantine rate is 2.0%, the cumulative number of cases reaches 2000 in 1 year and the cumulative number of quarantined infectives reaches 150 in 1 year. In the case that exposed infectives are symptomatic at day 7 (infectious 2 days before symptoms) and the maximum quarantine rate is 2.0%, the cumulative number of cases is approximately 3,700,000 and the cumulative number of quarantined infectives reaches 300,000 in 1 year (see Figure 7).

Exposed are infectious 1 day before symptoms

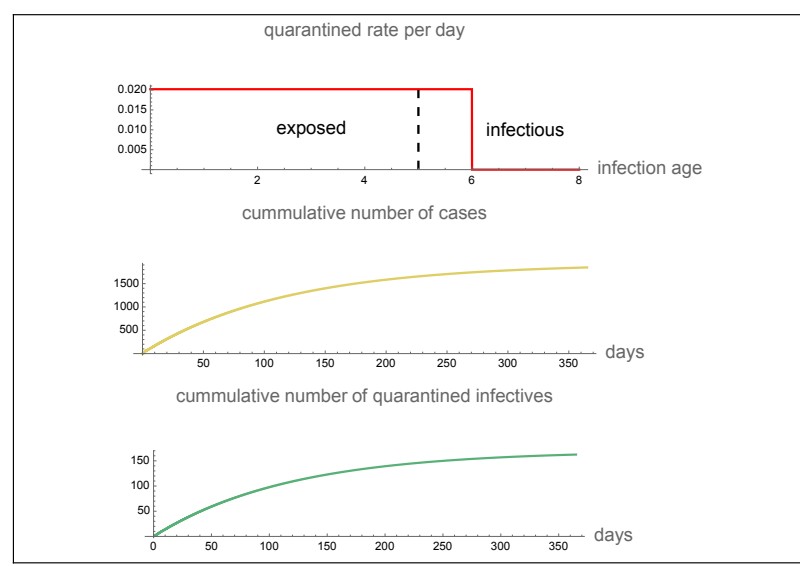

Exposed are infectious 2 days before symptoms

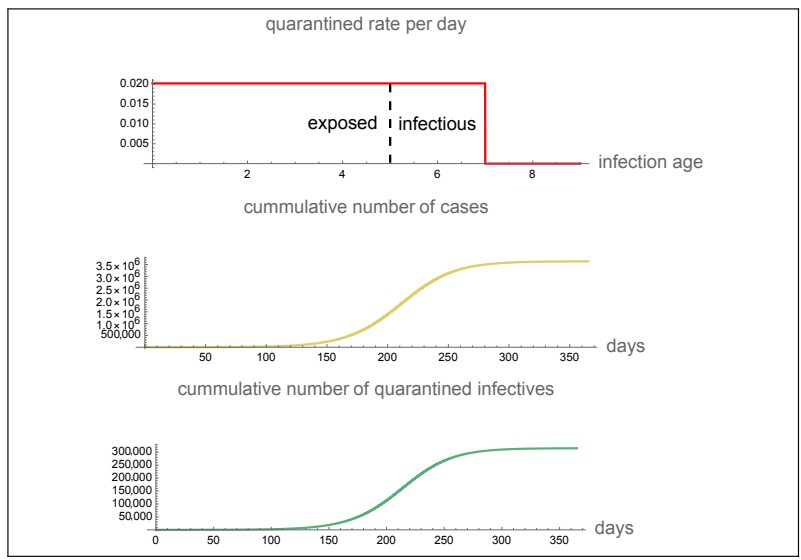

**Figure 7.** The epidemic in the case that (**top**) infectiousness precedes symptom on-set by 1 day and the maximum quarantine rate is 2.0% and (**bottom**) infectiousness precedes symptom on-set by 2 days and the maximum quarantine rate is 2.0%. Vertical dashed lines separate exposed and infectious. Red vertical lines represent the beginning of symptoms and the maximum disease age of quarantine.

In the case that exposed infectives are symptomatic at day 6 (infectious 1 day before symptoms) and the maximum quarantine rate is 4.0%, the cumulative number of cases reaches approximately 800 in 150 days and the cumulative number of quarantined infectives reaches 120 in 150 days. In the case that exposed infectives are symptomatic at day 7 (infectious 2 days before symptoms) and the maximum quarantine rate is 4.0%, the cumulative number of cases is approximately 2,700,000 in 450 days and the cumulative number of quarantined infectives reaches 400,000 in 450 days (see Figure 8).

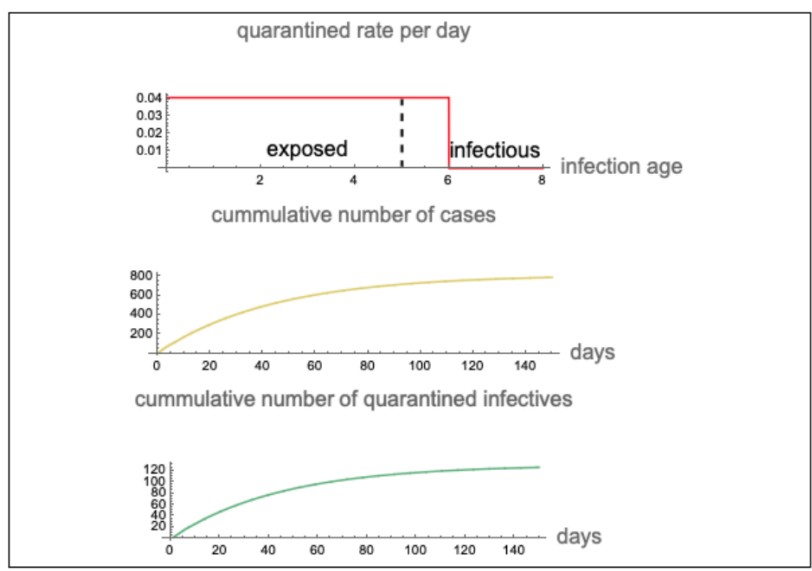

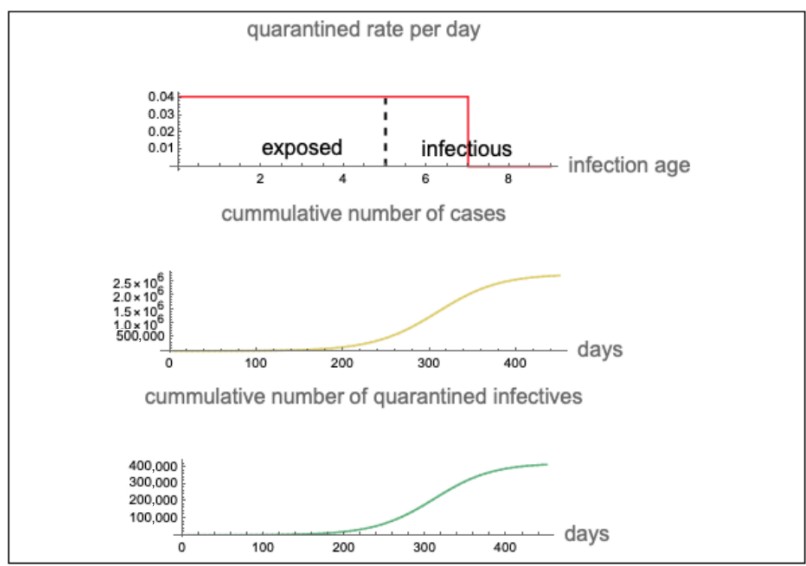

**Figure 8.** The epidemic in the case that (**top**) infectiousness precedes symptom on-set by 1 day and the maximum quarantine rate is 4.0% and (**bottom**) infectiousness precedes symptom on-set by 2 days and the maximum quarantine rate is 4.0%. Vertical dashed lines separate exposed and infectious. Red vertical lines represent the beginning of symptoms and the maximum disease age of quarantine.

In the case that exposed infectives are symptomatic at day 6 (infectious 1 day before symptoms) and the maximum quarantine rate is 10.0%, the cumulative number of cases reaches approximately 350 in 50 days and the cumulative number of quarantined infectives is approximately 100 in 50 days. In the case that exposed infectives are symptomatic at day 7 (infectious 2 days before symptoms) and the maximum quarantine rate is 10.0%, the cumulative number of cases is approximately 1000 in 200 days and the cumulative number of quarantined infectives reaches approximately 300 in 200 days (see Figure 9).

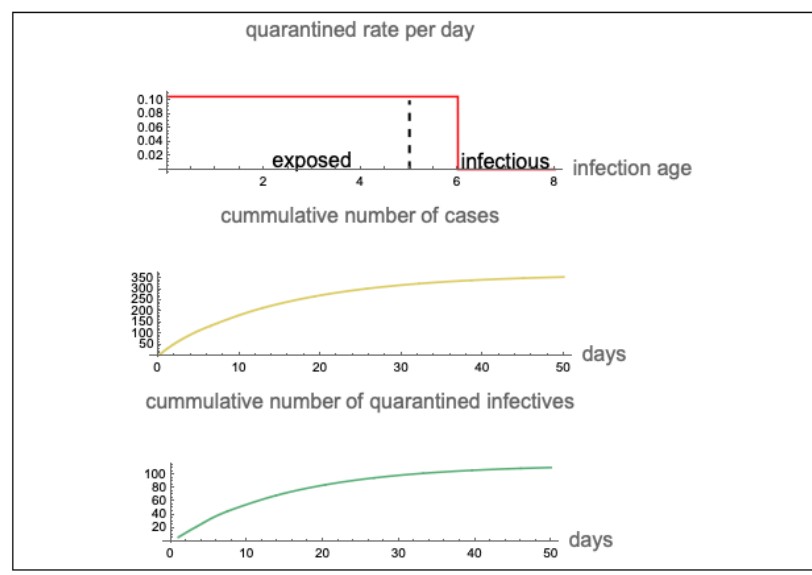

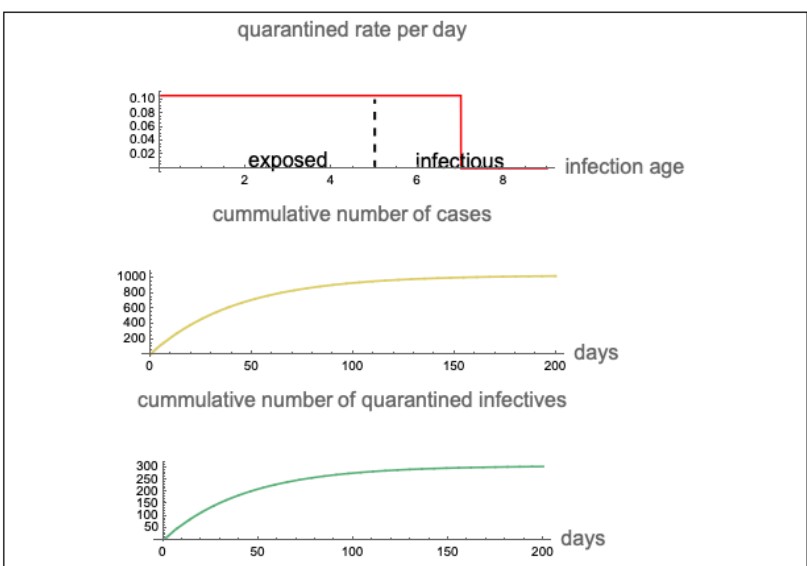

**Figure 9.** The epidemic in the case that (**top**) infectiousness precedes symptom on-set by 1 day and the maximum quarantine rate is 10.0% and (**bottom**) infectiousness precedes symptom on-set by 2 days and the maximum quarantine rate is 10.0%. Vertical dashed lines separate exposed and infectious. Red vertical lines represent the beginning of symptoms and the maximum disease age of quarantine.

In Figure 10, the graph of the total number of cases $\int_0^\infty i(0,t)dt$ as a function of the number of days of infectiousness pre-symptomatic $p$ and the maximum quarantine rate is given. The number of cases rises sharply as the quarantine rate falls below 5% and the number of days infectious pre-symptomatic exceeds 1.

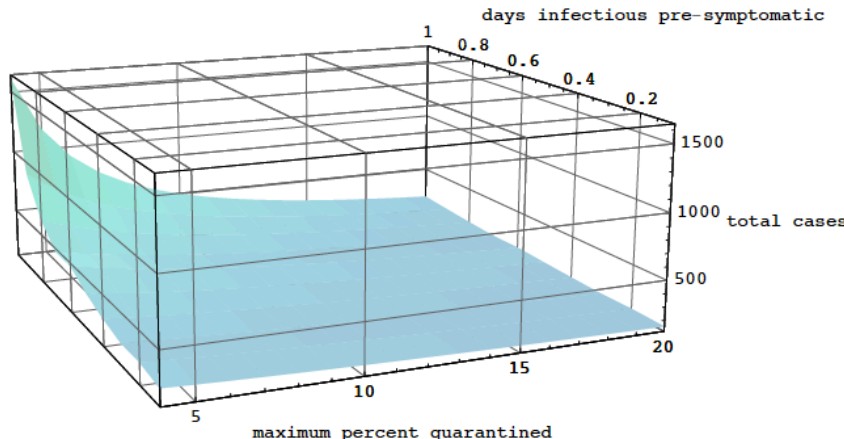

**Figure 10.** The total number of cases graphed as a function of the number of days infectiousness pre-symptomatic and the maximum quarantine rate.

We modify the model of the 2003 Taiwan SARS epidemic without vaccination to include vaccination, which was not available in Taiwan in 2003. This example will illustrate the epidemic evolution with alternate elements, including vaccination. We take the vaccination parameter $\nu = 0.05$. Vaccinated individuals begin with vaccinated age $a_v = 0$ and then acquire increasing immunity as their vaccination age increases over a period of days or weeks. The total number of vaccinated at time $t$ is $\int_0^t v(a_v, t)da_v$. Susceptibles are vaccinated at a constant rate $\nu$ per day. The proportion of vaccinated still susceptible at vaccination age $a_v$ is $\sigma(a_v)$.

In this example, $\nu = 0.05$, $\sigma(a_v) = 0.7e^{-0.25a_v} + 0.3$ (which means vaccination results in incomplete immunity, and as vaccination age $a_v$ advances, 30% of vaccinated individuals remain susceptible). We assume, $V(0) = 0$. Infectiousness precedes symptom onset by 2 days and the maximum quarantine rate is 4.0%. All other parameters are as before. The evolution of the epidemic is graphed in Figure 11, where it is seen that the cumulative number of cases is approximately 175,000. This example can be compared to the model with the same parameters, except without vaccination in the bottom graph in Figure 8, where the cumulative number of cases is approximately 350,000.

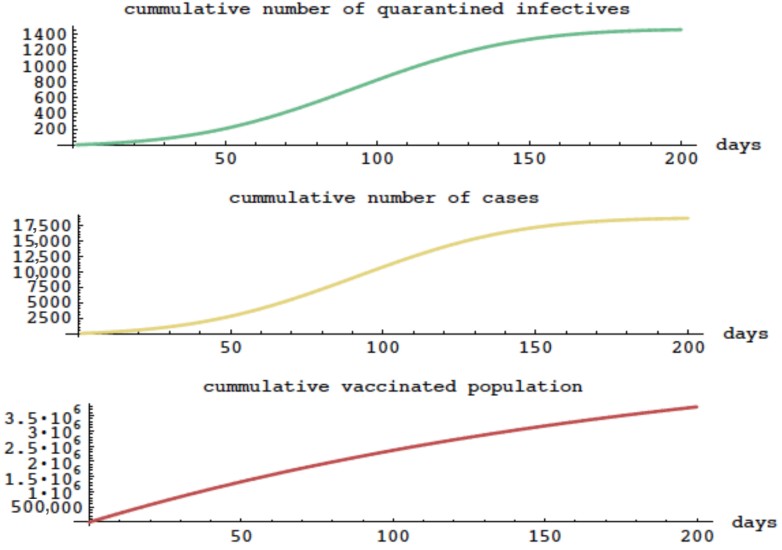

**Figure 11.** The cumulative number of quarantined cases, total number of cases, and cumulative number of vaccinated susceptibles in the model with vaccination.

*2.5. Application of the Model to the COVID-19 Epidemic in New York*

In this section, we apply our mathematical model to analyze the transmission dynamics of COVID-19 in New York. Numerous factors influence COVID-19 transmission, including vaccination rates, the emergence of more contagious variants, the public's reaction to and understanding of the virus, and governmental responses and policies. To provide a more detailed analysis, we segment the data into different phases, aligned with the timeline of COVID-19 transmission and the New York state government's response [118].

We obtain data from the New York State Department of Health (https://health.data.ny.gov/, accessed on 25 December 2023). The state of New York confirmed its first case of COVID-19 during the pandemic on 1 March 2020, while the first complete vaccination (i.e., two-dose vaccination) began on 15 December 2020. Our analysis focuses on the timeframe from 30 October 2020 to 13 March 2022.

In Figure 12, the green dots depict daily reported cases. Since these data tend to be erratic and are subject to ongoing updates, a standard approach is to use a rolling weekly average. Accordingly, the gray bars in the figure represent this rolling weekly average. The top figure in Figure 13 follows a similar presentation: green dots for daily vaccinated individuals and gray bars for the rolling weekly averages.

On average, symptoms of COVID-19 manifest in newly infected individuals approximately 5–6 days later (*WebMD*, https://www.webmd.com/covid/coronavirus-incubation-period, accessed on 15 October 2023) and last for about two weeks. We set the minimum age of infectiousness $r = 3$, the number of days of pre-symptomatic infectiousness $p = 2$, the number of days when symptoms appear $r + p = 5$, and the number of days of infectiousness $s = 11$. It is assumed that the hospitalization rate $\beta_H(a_i)$ per day is 54.5% once symptoms appear (after day $r + p = 5$), with a rate of 0.0% per day before day 5 (Figure 14):

$$\beta_H(a_i) = \begin{cases} 0 & \text{if } 0 \leq a_i < r + p, \\ 0.545 & \text{if } r + p \leq a_i \leq 14. \end{cases}$$

We assume only pre-symptomatic infected individuals are quarantined. The quarantine rate is 4.0% per day from day 0 to day 5 and then 0.0% per day after day 5 (see Figure 14):

$$\beta_Q(a_i) = \begin{cases} 0.020 & \text{if } 0 \leq a_i < r + p, \\ 0 & \text{if } r + p \leq a_i \leq 14. \end{cases}$$

In the case of COVID-19, the infectious period precedes the symptomatic phase. Individuals with COVID-19 can transmit the virus up to 48 h before they begin to show symptoms. Based on this understanding, we assume the exposed period for COVID-19 spans from the moment of infection to day 3 (i.e., $r = 3$). The infectious period then continues from day 3 to day 14, resulting in a two-day overlap between the pre-symptomatic and infectious periods (see Figure 15).

As mentioned, the transmission dynamics of COVID-19 are influenced by numerous factors, leading to multiple waves of infection as evidenced by the empirical data presented in Figure 12. In order to accurately model these multiple waves observed in the COVID-19 data, we have refined our simulation strategy. We segment the entire timeframe into different phases, assigning distinct transmission rates to each phase to reflect the changing epidemiological and social responses during the pandemic. The model is applied iteratively with these phase-specific transmission parameters, enabling our simulations to capture the multi-wave nature of the outbreak.

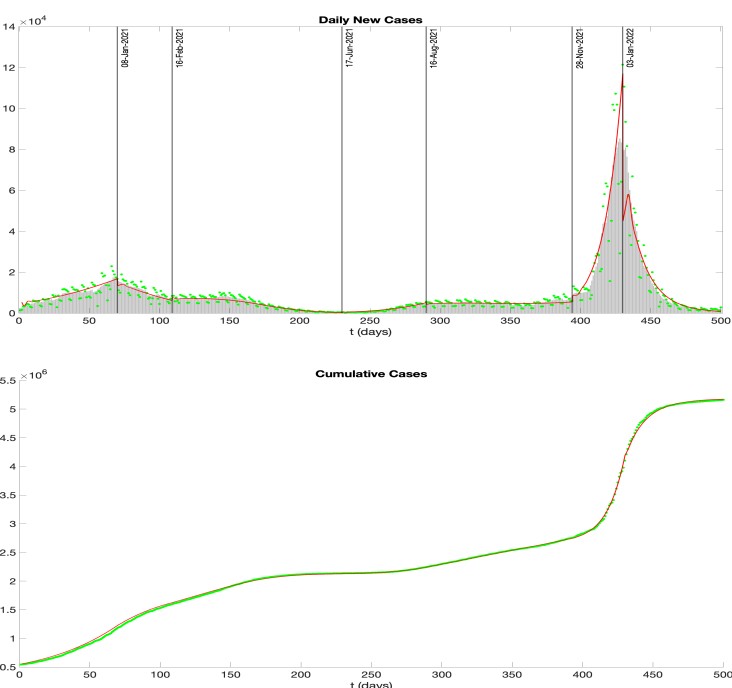

**Figure 12.** Daily and cumulative infectious cases for COVID-19 in New York from 30 October 2020 to 13 March 2022. (**Top**) Green dots represent data sourced from the New York State Department of Health (https://health.data.ny.gov/Health/New-York-State-Statewide-COVID-19-Testing/jvfi-ffup, accessed on 20 December 2023), gray bars show the rolling weekly averages, and the red curve is the simulation result of our model. (**Bottom**) Green dots represent data, and the red curve is the simulation result of our model.

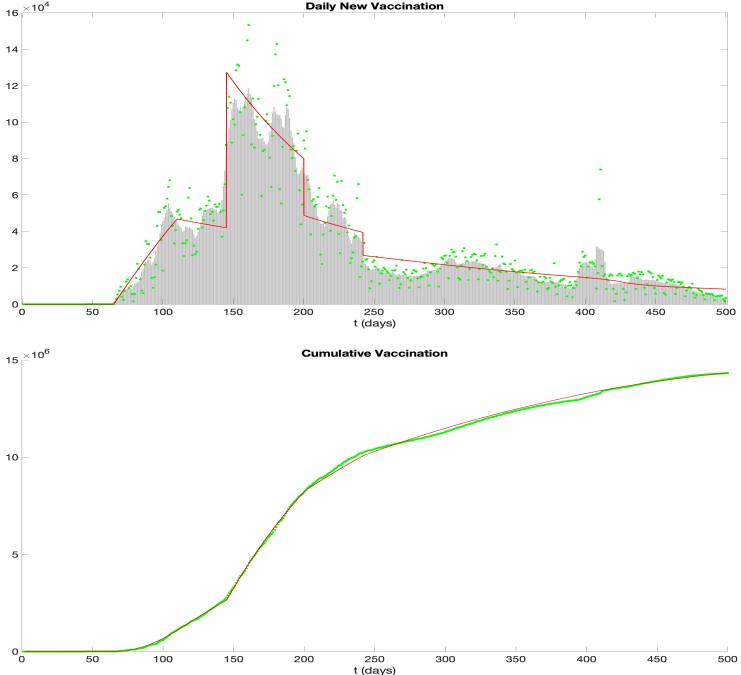

**Figure 13.** Daily and cumulative COVID-19 vaccinations in New York from 30 October 2020 to 13 March 2022. (**Top**) Green dots represent data from the New York State Department of Health (https://health.data.ny.gov/Health/New-York-State-Statewide-COVID-19-Vaccination-Data/duk7-xrni, accessed on 20 December 2023), gray bars represent the rolling weekly average, and the red curve is the simulation result of our model. (**Bottom**) The green dotted curve represents the data, and the red curve illustrates the outcomes from our model simulation.

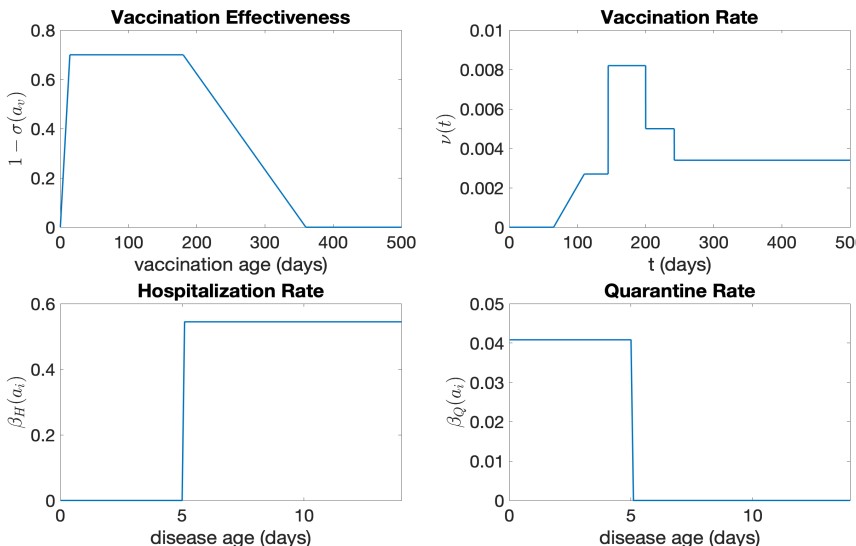

**Figure 14.** The vaccination age-dependent effectiveness, vaccination rate, hospitalization rate, and quarantine rate for the COVID-19 epidemic in New York from 30 October 2020 to 13 March 2022.

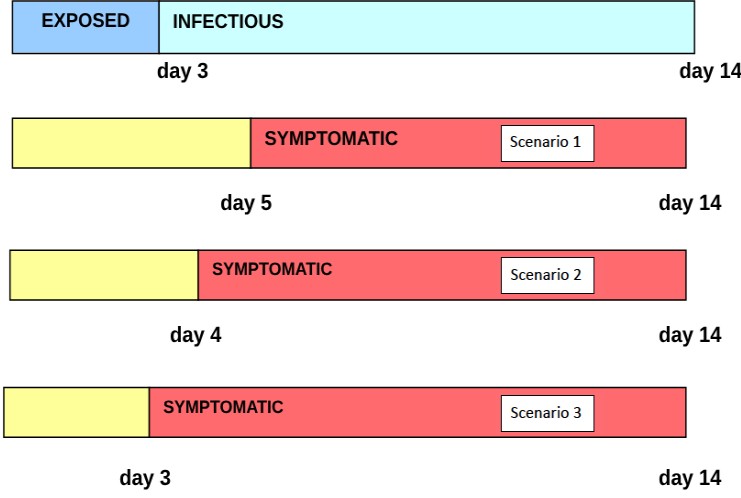

**Figure 15.** Timeline of infectious periods relative to symptom onset for COVID-19. The top segment displays the exposed–infectious period. Segments 2 to 4 illustrate scenarios where the infectious period starts two days before, one day before, and simultaneously with symptom onset, respectively.

Upon fitting the data, we have different values of transmission rates in different phases:

- Phase 1: 1 November 2020 to 8 January 2021. There was no vaccination in this phase.

$$
\alpha(a_i) = \begin{cases} 0 & \text{if } 0 \le a_i < r, \\ 8.4 \times 10^{-9}(a_i - r) & \text{if } r \le a_i < r+7, \\ 8.4 \times 10^{-9} \times \frac{7}{s-7}(r+s-a_i) & \text{if } r+7 \le a_i < r+s, \\ 0 & \text{if } r+s \le a_i. \end{cases}
$$

- Phase 2: 8 January 2021 to 16 February 2021. With the commencement of vaccination campaigns and growing public caution, there was a small decrease in the COVID-19 transmission rate.

$$\alpha(a_i) = \begin{cases} 0 & \text{if } 0 \le a_i < r, \\ 6.7 \times 10^{-9}(a_i - r) & \text{if } r \le a_i < r + 7, \\ 6.7 \times 10^{-9} \times \frac{7}{s-7}(r + s - a_i) & \text{if } r + 7 \le a_i < r + s, \\ 0 & \text{if } r + s \le a_i. \end{cases}$$

- Phase 3: 16 February 2021 to 17 June 2021. The emergence and prevalence of the Alpha variant [119] brought a small increase in the transmission rate.

$$\alpha(a_i) = \begin{cases} 0 & \text{if } 0 \le a_i < r, \\ 8.3 \times 10^{-9}(a_i - r) & \text{if } r \le a_i < r + 7, \\ 8.3 \times 10^{-9} \times \frac{7}{s-7}(r + s - a_i) & \text{if } r + 7 \le a_i < r + s, \\ 0 & \text{if } r + s \le a_i. \end{cases}$$

- Phase 4: 17 June 2021 to 16 August 2021. In June 2021, the arrival of the Delta variant [120] led to a rapid surge in COVID-19 cases. It is estimated that the Delta variant is 60%–90% more transmissible than the Alpha variant [120,121].

$$\alpha(a_i) = \begin{cases} 0 & \text{if } 0 \le a_i < r, \\ 17.7 \times 10^{-9}(a_i - r) & \text{if } r \le a_i < r + 7, \\ 17.7 \times 10^{-9} \times \frac{7}{s-7}(r + s - a_i) & \text{if } r + 7 \le a_i < r + s, \\ 0 & \text{if } r + s \le a_i. \end{cases}$$

- Phase 5: 16 August 2021 to 28 November 2021. In response to the rise in the Delta variant in August 2021, policies such as a universal mask mandate for all public and private schools were implemented [118], leading to a reduced transmission rate.

$$\alpha(a_i) = \begin{cases} 0 & \text{if } 0 \le a_i < r, \\ 14.2 \times 10^{-9}(a_i - r) & \text{if } r \le a_i < r + 7, \\ 14.2 \times 10^{-9} \times \frac{7}{s-7}(r + s - a_i) & \text{if } r + 7 \le a_i < r + s, \\ 0 & \text{if } r + s \le a_i. \end{cases}$$

- Phase 6: 28 November 2021 to 3 January 2022. The Omicron variant [122] was first discovered in Botswana and South Africa in November 2021 and quickly spread to other countries, including the United States. In December 2021, the emergence of the Omicron variant led to a significant surge in COVID-19 cases.

$$\alpha(a_i) = \begin{cases} 0 & \text{if } 0 \le a_i < r, \\ 23.6 \times 10^{-9}(a_i - r) & \text{if } r \le a_i < r + 7, \\ 23.6 \times 10^{-9} \times \frac{7}{s-7}(r + s - a_i) & \text{if } r + 7 \le a_i < r + s, \\ 0 & \text{if } r + s \le a_i. \end{cases}$$

- Phase 7: 3 January 2022 to 13 March 2022. Reacting to the emergence of the Omicron variant, various preventive policies, such as mask mandates and "Comprehensive Winter Surge Plans" were introduced [118], leading to a decrease in the transmission rate.

$$\alpha(a_i) = \begin{cases} 0 & \text{if } 0 \leq a_i < r, \\ 8.7 \times 10^{-9}(a_i - r) & \text{if } r \leq a_i < r + 7, \\ 8.7 \times 10^{-9} \times \frac{7}{s-7}(r + s - a_i) & \text{if } r + 7 \leq a_i < r + s, \\ 0 & \text{if } r + s \leq a_i. \end{cases}$$

The transmission rate $\alpha(a_i, t)$ as a function of disease age $a_i$ and time $t$ is depicted in Figure 16.

Similar to the example of SARS, we utilize the daily reported data from 16 October 2020 to 29 October 2020 (a fourteen-day period before 30 October 2020) as the initial distribution of $i(a_i, 0)$. Specifically, at $t = 0$:

$$i(a_i, 0) = \begin{cases} 2328 & \text{if } a_i \leq 1, \\ 2369 & \text{if } 1 < a_i \leq 2, \\ 2511 & \text{if } 2 < a_i \leq 3, \\ 2328 & \text{if } 3 < a_i \leq 4, \\ 2314 & \text{if } 4 < a_i \leq 5, \\ 1164 & \text{if } 5 < a_i \leq 6, \\ 1304 & \text{if } 6 < a_i \leq 7, \\ 2028 & \text{if } 7 < a_i \leq 8, \\ 2337 & \text{if } 8 < a_i \leq 9, \\ 2177 & \text{if } 9 < a_i \leq 10, \\ 2044 & \text{if } 10 < a_i \leq 11, \\ 2177 & \text{if } 11 < a_i \leq 12, \\ 1189 & \text{if } 12 < a_i \leq 13, \\ 1115 & \text{if } 13 < a_i \leq 14. \end{cases}$$

The graph of this initial disease age distribution $i(a_i, 0)$ is plotted in Figure 14. We assume $S(0) = 19,500,000$ (https://usafacts.org/data/topics/people-society/population-and-demographics/, accessed on 15 October 2023).

In contrast to the SARS outbreak in Taiwan in 2003, where vaccination was not an option, the availability and administration of COVID-19 vaccines have significantly influenced the dynamics of its transmission. While COVID-19 vaccines have proven to offer substantial protection to those who are susceptible, they are not infallible—people can still be infected with COVID-19 after vaccination. This means that the COVID-19 vaccination is not 100% effective. We assume that the vaccination age-dependent function $\sigma(a_v)$ decreases from 1 to 0.3 within two weeks, resulting in a 70% effectiveness for COVID-19 vaccines. This level of effectiveness persists for six months and then steadily wanes, reaching 0% (i.e., $1 - \sigma = 0$) after a year. This assumption is based on the administration of annual boosters, indicating that the COVID-19 vaccines' protection wanes after a year. The graph of $1 - \sigma(a_v)$ is shown in Figure 14.

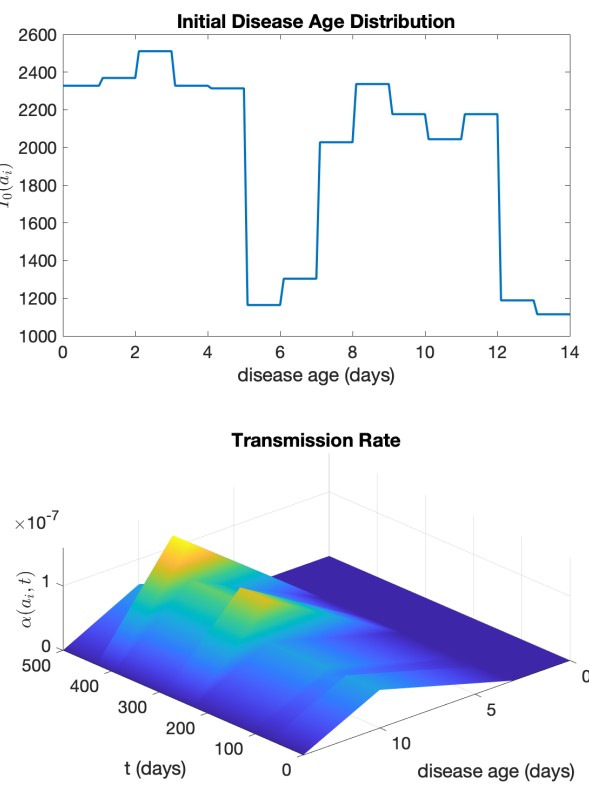

**Figure 16.** (**Top**) The initial disease age distribution and (**bottom**) the phase-dependent disease transmission rate, as a function of the disease age $a_i$ and time $t$, for the COVID-19 epidemic in New York from 30 October 2020 to 13 March 2022.

The vaccination rate $\nu(t)$ (see Figure 14) is fitted using the daily vaccination data for New York (https://health.data.ny.gov/Health/New-York-State-Statewide-COVID-19-Vaccination-Data/duk7-xrni, accessed on 20 December 2023). It takes the form

$$
\nu(t) = \begin{cases}
0 & \text{if } t \leq 65, \\
6 \times 10^{-5}(t-65) & \text{if } 65 < t \leq 110, \\
0.27\% & \text{if } 110 < t \leq 145, \\
0.82\% & \text{if } 145 < t \leq 200, \\
0.50\% & \text{if } 200 < t \leq 242, \\
0.33\% & \text{if } t > 242.
\end{cases}
$$

We assume a vaccination rate of 0 before $t = 65$, aligning with the actual start of complete vaccinations (i.e., two-dose vaccination) in New York on 15 December 2020. The rising vaccination rate from $t = 65$ to $t = 110$ reflects the initial scarcity of vaccine doses, which were prioritized for older adults and high-risk hospital workers. As vaccine production surged and more vaccination sites were established, the pace of vaccinations increased, thus increasing vaccination rates. After $t = 110$, we assume a series of distinct constant vaccination rates, each applicable to specific time intervals, to best represent the varying pace of vaccination during those periods. These constant rates for each interval have been determined based on data fitting.

We employ the Forward Euler Scheme with a time step of 0.1 to discretize our model using the parameter values mentioned above. The resulting graph depicting the daily and cumulative infections is represented by the red curves in Figure 12. It agrees well with

the data, and our simulated curve aptly captures the significant surge in COVID-19 cases attributed to the Omicron variant.

We analyze the effects of $p$ on the number of infectives in the model. The parameter $p$ represents the number of days during which the infectious and pre-symptomatic periods overlap. We explore two scenarios: $p = 1$, where the infectious period precedes symptoms by a day (see Scenario 2 in Figure 15); and $p = 0$, where the infectious period and the symptomatic period coincide (see Scenario 3 in Figure 15). With other parameters held constant, the results for daily new and cumulative infectious cases are illustrated in Figures 17 and 18. Notably, for $p = 0, 1$, daily new cases near zero appear after 50 days, indicating effective disease control. This underscores the efficacy of hospitalizing symptomatic patients as a means to isolate infectious individuals and control the disease's progression.

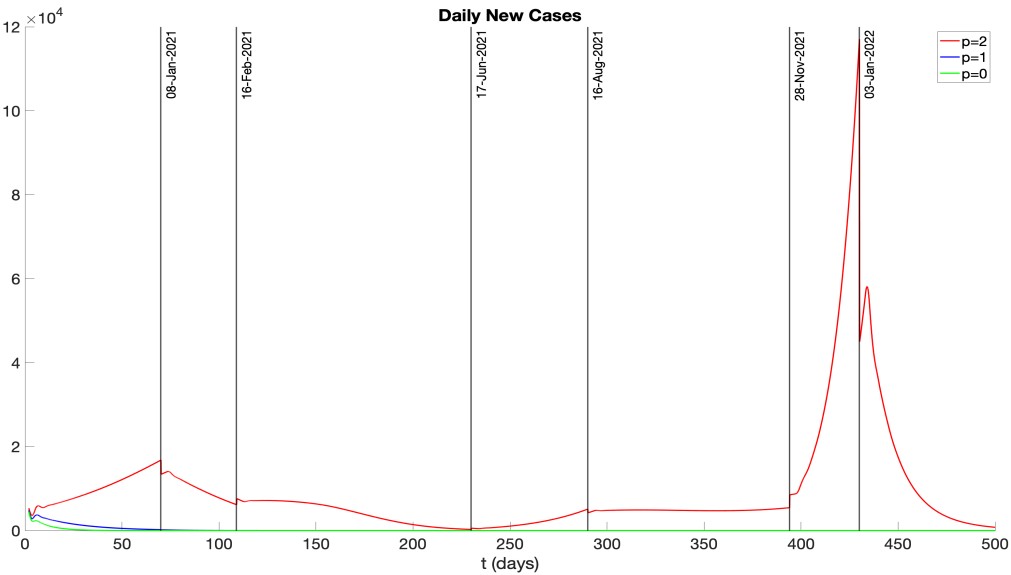

**Figure 17.** Daily new infectious cases represented by different curves for $p$ values of 2, 1, and 0. Here, $p$ denotes the overlap in days between infectious and pre-symptomatic periods.

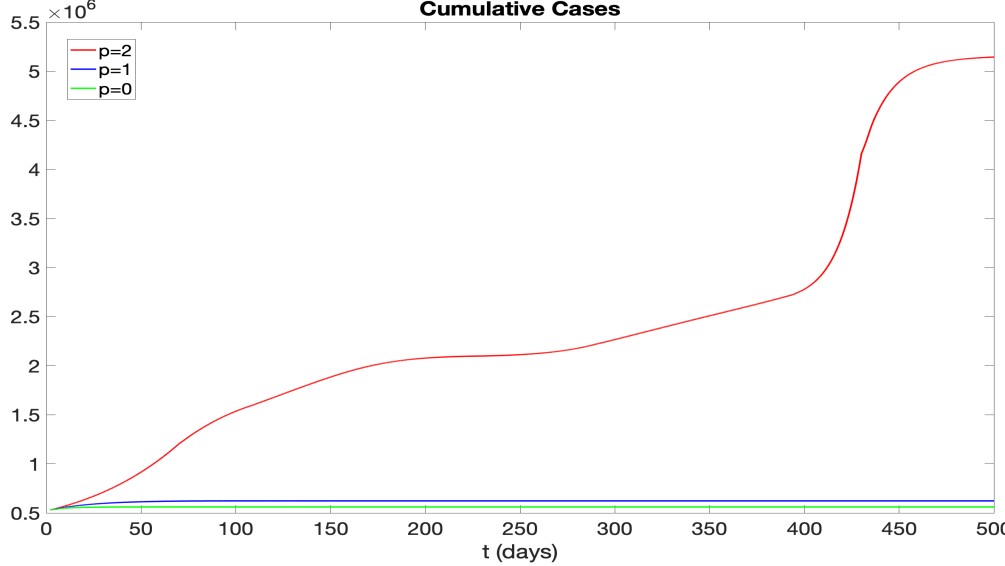

**Figure 18.** Cumulative infectious cases represented by different curves for $p$ values of 2, 1, and 0. Here, $p$ denotes the overlap in days between infectious and pre-symptomatic periods.

In addition, we further examine the effects of varying vaccination rates on the number of infectives. Specifically, we consider two scenarios: one with a vaccination rate of $0.5\nu(t)$

and another with $2\nu(t)$. Figures 19 and 20 display the daily new and cumulative infectious cases for vaccination rates of $\nu(t)$, $0.5\nu(t)$, and $2\nu(t)$. Our findings suggest that a lower vaccination rate results in a higher number of infectious cases. Moreover, if the vaccination rate is doubled in the initial stage, the disease can be fully suppressed by approximately day 200. Another noteworthy observation is that with a vaccination rate of $0.5\nu(t)$, there is a peak in daily new infections around day 300. Yet, during the phase attributed to the Omicron variant, the number of new infections is significantly lower. This can be attributed to our assumption that infectives are not susceptible to re-infection. Consequently, the peak of infections around day 300 significantly reduces the number of susceptible individuals, and thus there are not many new infectious cases after day 400.

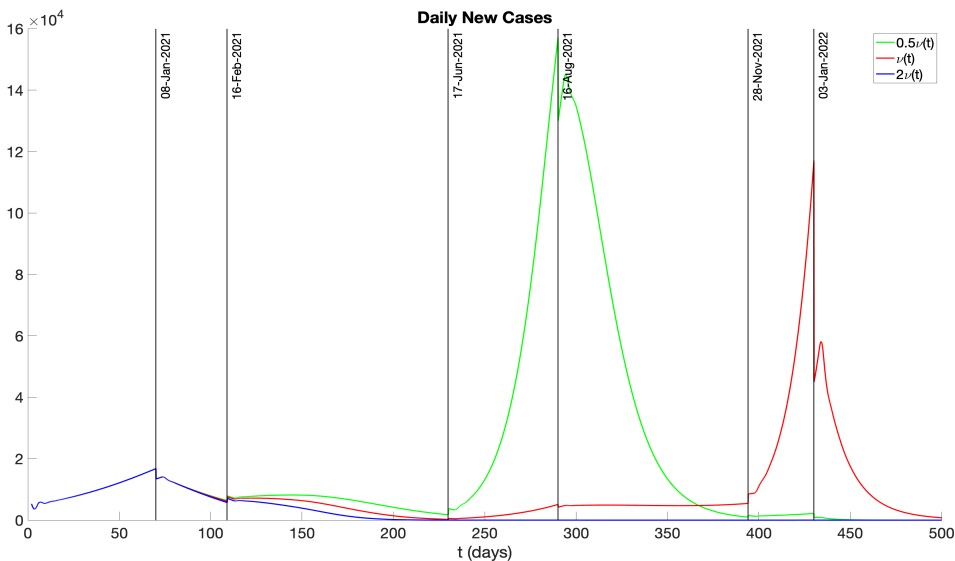

**Figure 19.** Daily new infectious cases with different vaccination rates $\nu(t)$, $2\nu(t)$, and $0.5\nu(t)$.

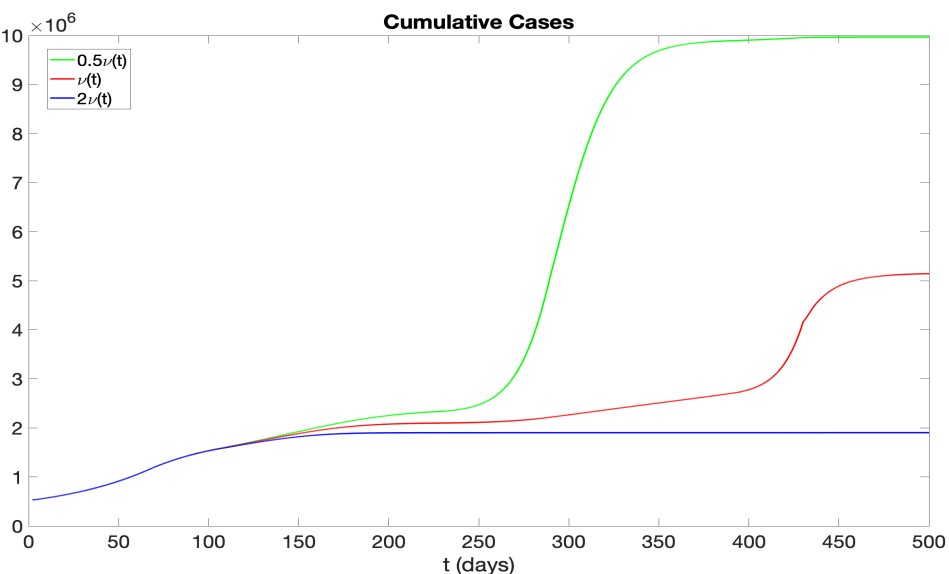

**Figure 20.** Cumulative infectious cases with different vaccination rates $\nu(t)$, $2\nu(t)$, and $0.5\nu(t)$.

Building on our analysis of vaccination rates, we next turn our attention to the role of hospitalization rates in controlling the spread of COVID-19. We investigate how variations in the hospitalization rate (after symptoms appear), denoted by $\beta_H$, affect the number of infections. In Figure 21, we present the daily new cases, and in Figure 22, we illustrate the cumulative cases, each for a range of values from 0.51 to 0.58. The different curves

in these figures demonstrate the sensitivity of the infection dynamics to hospitalization practices, revealing that higher hospitalization rates can significantly flatten the curve and reduce the total number of infections over time. These insights point to the critical impact of hospitalization rates on the management of the disease, alongside vaccination strategies.

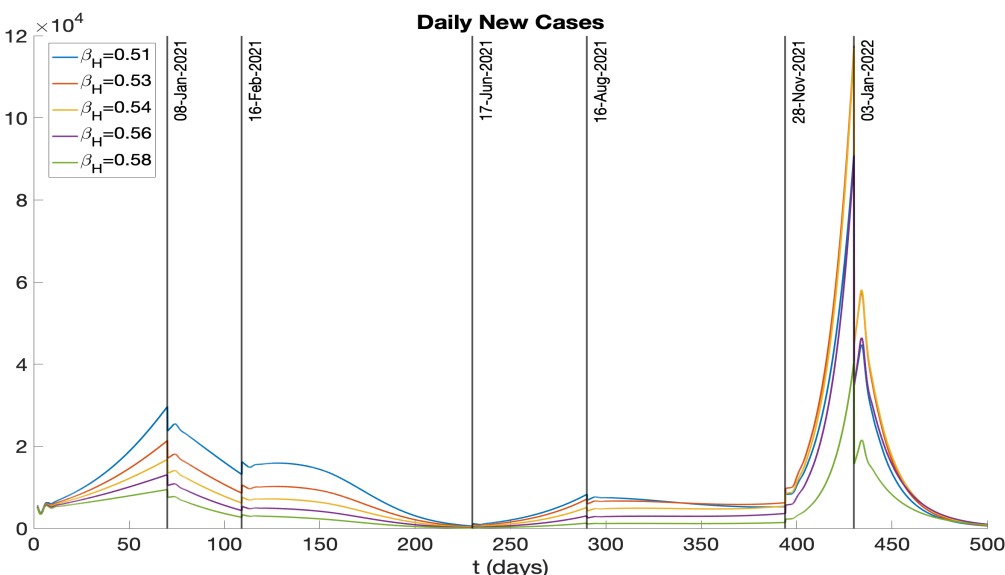

**Figure 21.** Daily new infectious cases represented by different curves for different $\beta_H$ values.

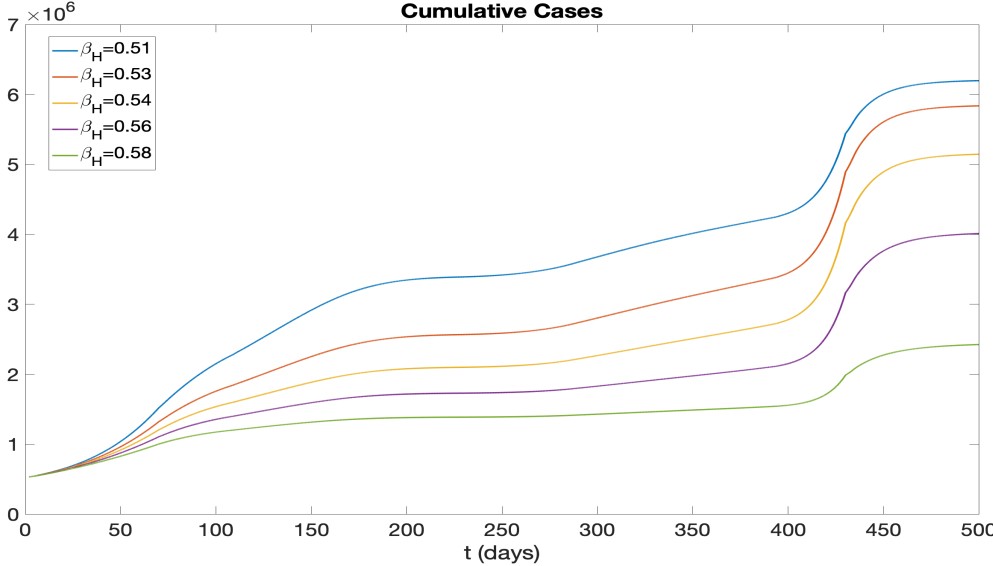

**Figure 22.** Cumulative infectious cases represented by different curves for different $\beta_H$ values.

## 3. Conclusions

We have developed an epidemic model structured by infection age and vaccination age. The epidemic dynamics are analyzed with respect to the infection age of infected individuals before symptoms appear, the fraction of pre-symptomatic infected individuals placed in quarantine, the hospitalization rate of symptomatic infectives, and the vaccine efficacy of vaccinated individuals with respect to their vaccination age. The model is numerically simulated for the 2003 SARS epidemic in Taiwan (data from Taiwan Centers for Disease Control) and the COVID-19 epidemic in the state of New York (data from New York State Department of Health). The parameters for the model simulations are fitted to these data based on weekly rolling averages of the daily data in these data sources.

The computer codes for our numerical simulations are available upon request in both MATHEMATICA and MatLab.

In the application of the model to the 2003 SARS epidemic in Taiwan, in which vaccination was not available, the on-set of symptoms and the beginning of the infectious period coincided. The quarantine rate of susceptibles was approximately 2% per day. The epidemic was contained with the maximum cumulative number of infected individuals at approximately 230 in 100 days (Figure 1).

We modified the pre-symptomatic and infectious periods to $p = 1$ and $p = 2$ days of infectious pre-symptomatic cases. We also modified the quarantine rate to 4% and 10%. The results are shown in Figures 7–9, where it is seen that the value $p = 2$ has much higher cumulative cases than the value $p = 1$. When $p = 2$, the quarantine rate must be very high to significantly reduce the cumulative number of cases.

We also modified the 2003 SARS epidemic model to illustrate the impact of vaccination. In Figure 10, it is seen that infectious 2 days pre-symptomatic ($p = 2$), quarantine rate 4%, and the vaccine efficacy $\sigma(a_v) = 0.7 \exp(-0.25 a_v) + 0.3$, result in a reduction in the maximum cumulative number of cases to $175,000$ compared to $350,000$ without vaccination.

In the application of the model to the COVID-19 epidemic in New York, the infectious period is pre-symptomatic by $p = 2$ days. This pre-symptomatic infectious period is a key feature of COVID-19 epidemic dynamics. From data sources, we parameterized the model into seven phases corresponding to vaccine implementation, viral variants, and social responses. The model simulations agree with the observed infection and vaccination (Figures 12 and 13).

We examined the consequences of modifying the pre-symptomatic infectious period, initially set at $p = 2$ days, by considering $p = 1$ day and $p = 0$ days. Figures 17 and 18 display numerical simulations for $p = 0, 1, 2$. It is seen that $p = 0$ and $p = 1$ result in a major reduction in the epidemic impact. We also investigated the impact of varying the vaccination rate parameter $\nu(t)$. Figures 19 and 20 show that an increase in $\nu(t)$ significantly mitigates the epidemic, while a decrease in $\nu(t)$ exacerbates the epidemic. Furthermore, we explored the effects of changing the hospitalization parameter $\beta_H(a_i)$. In Figures 21 and 22, we observed that higher values of $\beta_H(a_i)$ significantly decrease the epidemic's severity, whereas lower values of $\beta_H(a_i)$ increase the epidemic's severity.

In general, incorporating infection age and vaccination age into our analysis enables a detailed examination of key factors affecting epidemic outcomes. The continuum formulation of the infection age and vaccination age provides applicable parameter identification and numerical simulation of this age structure. Specifically, infection age and vaccination age can be connected to critical elements, such as pre-symptomatic infectiousness, vaccination efficacy, and hospitalization rate, which are integral to understanding and predicting epidemic dynamics.

While our model incorporates various factors, it remains a simplified representation of real-world disease transmission. It is important to highlight potential areas for refinement to make the model more realistic. For instance, reinfections are notably common with COVID-19 [123]. As the Omicron variant became predominant, data indicated a significant rise in reinfection rates among all COVID-19 cases [124]. Additionally, revaccination is another factor to account for, given the CDC's recommendation for individuals aged 12 and older to receive an updated COVID-19 vaccine annually [125]. We will consider models for this extension in our future study. Notably, another promising direction for enhancement is the integration of chronological age. Data on COVID-19 are often categorized by age groups, and different age brackets might exhibit varied transmission rates [126–128]. This consideration will be a focus in our subsequent analyses.

**Author Contributions:** Conceptualization, G.W. and X.E.Z.; Methodology, X.E.Z.; Validation, G.W.; Formal analysis, G.W. and X.E.Z.; Investigation, G.W. and X.E.Z.; Data curation, G.W.; Writing—original draft, G.W. and X.E.Z. All authors have read and agreed to the published version of the manuscript.

**Funding:** This research received no external funding.

**Institutional Review Board Statement:** Not applicable.

**Informed Consent Statement:** Not applicable.

**Data Availability Statement:** The data on COVID-19 transmission used in this study are publicly available on the New York State Department of Health webpages. The computer codes developed for the numerical simulations in MATHEMATICA and MatLab are available upon request from the corresponding author.

**Conflicts of Interest:** The author declares no conflicts of interest.

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
