# Peer review of "An Epidemic Model with Infection Age and Vaccination Age Structure"

_2036-7449, doi:10.3390/idr16010004_

Round 1

Reviewer 1 Report

Comments and Suggestions for Authors

I believe that this is a very interesting and important work.

On page 10 a formula is given for R_0 when there is no vaccination. I wonder how is this formula obtained? By the next generation approach or otherwise? It would be desirable to include some details for the interested reader.

Also, do you compute or approximate the value of R_0 (based on this formula or in a different way) later on when parametrizing the model?

What is the importance of the result in the Remark on page 6? Maybe you could add some details/notes to explain.

A minor comment is that in the Conclusions section on page 23, you say that the population dynamics are analyzed. However, birth and death processes are not considered (understandably). You could try to better emphasize the distinction between disease dynamics (which I believe is studied in here) and population dynamics.

Another minor comment is that you write: "the current
COVID-19 epidemic in New York State" in the abstract and "In the application of the model to the current COVID-19 epidemic in New York State"; in the Conclusions section. I believe/hope that the COVID-19 epidemic is over in New York state, ideed the data refers to 2020 October to 2022 March.

To improve the aesthetics it would be desirable to have a uniform format in the References section (I appreciate that it won't be easy given the large number). Note e.g. the difference between refs. 21-25 and the rest.

Reviewer 2 Report

Comments and Suggestions for Authors

In this paper, the Authors generalize the traditional SIR and SEIR models (and some of their extensions) used to study the development of diseases and epidemics. They additionally incorporated the age of infection and the age of vaccination into the model. They based their formal analysis on the concepts of infection age density and vaccination age density. The model was verified on data regarding the 2003 SARS epidemic in Taiwan and the COVID-19 epidemic in New York State.

The paper is well-written and easy to follow. I believe it should be published in the journal Infectious Disease Reports.

The citation should be corrected (title page p. 1); there is no second Author's name.

Reviewer 3 Report

Comments and Suggestions for Authors

The topic presented is interesting and the mathematical results are sound. The paper is well-written but there is a lot of room for improvement. As general suggestions, I would improve the Abstract since it should be larger and more informative and I would shorten the long list of references (like half of them). Moreover, I would suggest changing the title to something like “Continuously structured epidemic model with infection and vaccination ages” (If I’m not wrong, the paper deals with just one model with many parameters, applied to two situations Taiwan and NYS). The topic of infection transmission prior to symptom onset is a very important issue, but the paper should be improved to be published by the journal.

In the Abstract you should already highlight why is important to consider both infection age and vaccination age in the model.

In the introduction, what you mean by epidemic population?

Section 2: you should explain in words the meaning of each state variable.

In the diagram of Fig. 1, it is not clear the individuals who are pre-symptomatic and symptomatic. Can you explain it? The modeling part should be enhanced.

For a discrete-time model dealing with asymptomatic and symptomatic individuals, see Mathematics 2023, 11, 1092. https://doi.org/10.3390/math11051092

Consider removing Figure 2. In my opinion, it is unnecessary.

In page 10, you should say basic reproduction number (remove the word epidemic) and the definition is “The basic reproduction number R0 is the expected number of secondary cases generated by each primary case over the disease course.”, see Volume 384, 1 March 2021, 113165. https://www.sciencedirect.com/science/article/pii/S0377042720304568

In Figure 10, write basic reproduction number instead.

In Figure 13, write Scenario 1, Scenario 2 and Scenario 3 in the red bars.

The model presented (5) - (10) is an epidemic model for a single outbreak of an infectious disease. On the contrary, the data displayed in Figure 14 correspond to multiple waves (several outbreaks). You should explain this fact. 

In Conclusions section, it is not clear how model parameters are fitted to real data. Consider removing Figure 24.

As a general comment, I would say that the work done should be transformed from a Chapter in a a Ph.D. thesis to a publishable paper. The authors should simplify the exposition, in particular reduce the number of Figures. In my opinion, the modeling part is not well-addressed.

Comments on the Quality of English Language

Not qualified.

Round 2

Reviewer 3 Report

Comments and Suggestions for Authors

The new version of the manuscript is ready for publication by the journal. Are all the references of the paper cited in the main text?

Comments on the Quality of English Language

Not applicable.

Author Response

December 27, 2023
We have cited all our References in this revision. We believe that our general References [1-110] provide a valuable source for researchers investigating epidemic models with structure variables and vaccination elements. We thank Reviewer #3 for helpful comments.
Sincerely,
Glenn Webb     Evelyn Zhao
